# Association between optical coherence tomography and functionally severe stenosis assessed by quantitative flow ratio in coronary intermediate lesions

Yuan Xue[1], Yuchen Zhang[2], Peng Wu[3], Yang Yang[3], Xin Lu ![ORCID][4]*, Haoyu Meng[3]*

**1** Department of Cardiology, Gaoyou People's Hospital, Yangzhou, China, **2** Department of Cardiology, The Fouth Affiliated Hospital of Nanjing Medical University, Nanjing, China, **3** Department of Cardiology, The First Affiliated Hospital of Nanjing Medical University, Nanjing, China, **4** Department of Cardiology, Affiliated Hospital of Nanjing University of Chinese Medicine, Nanjing, China

* luxin1234@njucm.edu.cn (XL); drhymeng@njmu.edu.cn (HM)

## Abstract

### Background

Stenosis severity for an intermediate coronary lesion (ICL) demands comprehensive assessment methods. The relationship between optical coherence tomography (OCT) and coronary stenosis evaluated by quantitative flow ratio (QFR) remains to be discussed.

### Objective

To investigate the association between OCT parameters and coronary functionally significant stenosis identified by QFR in patients with ICL.

### Methods

A total of 113 patients with 115 target vessels were assessed using both OCT and QFR measurements for ICL. ICL was defined as a coronary lesion according to visual estimation with a diameter stenosis between 50% and 90%. Functional relevance stenosis was considered severe when QFR ≤ 0.80.

### Results

The mean age of the included patients was 62.2 years, and the overall median QFR was 0.82. The mean minimal lumen area (MLA) and area stenosis (AS) assessed by OCT were 3.11 mm$^2$ and 68.4%, respectively. According to the cut-off value of QFR = 0.80, the vessels with hemodynamic ischemia (n = 55) or without ischemia (n = 60) were grouped. OCT-measured MLA and AS exhibited a moderate diagnostic value with an area under the curve (AUC) of 0.83 (95% CI 0.75 to 0.90) for MLA and

**Data availability statement:** All relevant data are within the paper and its Supporting Information files.

**Funding:** National Natural Science Foundation of China (82400457), Natural Science Foundation of Jiangsu Province (BK20210966), and Clinical project of Jiangsu Provincial People's Hospital (303103513BA20). The funders had no role in study design, data collection and analysis, decision to publish, or preparation of the manuscript.

**Competing interests:** The authors have declared that no competing interests exist.

0.87 (95% CI 0.81 to 0.94) for AS, respectively. The optimal cutoff value exhibited by OCT-measured anatomical parameters for identifying coronary stenosis, represented by QFR ≤ 0.80 was 2.52 mm$^2$ (sensitivity 71%, specificity 85%) for MLA and 69% (sensitivity 84%, specificity 78%) for AS. Subsequently, AS measured by QFR and OCT were evaluated using linear regression analysis, revealing a good correlation (r = 0.79, p<0.001).

## Conclusions

Intraluminal anatomical parameters obtained by OCT showed moderate diagnostic value for predicting the functional ischemia evaluated by QFR. Additionally, QFR may offer a rapid and simple method to assess area stenosis in coronary intermediate lesions.

## Introduction

The comprehensive assessment of stenosis severity for intermediate coronary lesion (ICL) remains an ongoing challenge, requiring modalities to optimize clinical decision-making. It has been demonstrated that fractional flow reserve (FFR), as the physiology benchmark used in invasively assessing coronary stenosis, can serve as a guide for percutaneous coronary intervention (PCI), thus reducing adverse cardiac outcomes [1]. However, this kind of invasive operation requires an extended procedure time and may lead to complications in addition to its high cost limiting clinical application. Optical coherence tomography (OCT) employs light reflection to assist in quantifying the intraluminal dimensions and evaluating the atheromatous plaque in vivo [2]. OCT was introduced to provide additional morphological and anatomical information to guide percutaneous revascularization procedures. Several studies have focused on investigating the potential of OCT to help predict hemodynamically severe stenoses based on FFR [3–7]. Quantitative flow ratio (QFR) is a relatively new, low-cost and safe method used to evaluate the functional significance of coronary stenosis achieved through calculating the FFR in the vessel by using a 3D angiographic reconstruction algorithm together with a fluid dynamics algorithm. Several studies have shown that QFR has good diagnostic performance using fractional flow reserve (FFR) as its reference standard [8–10]. However, the relationship between coronary stenosis measured by OCT and ischemia assessed by QFR has yet to be investigated. This study centered on assessing the association between OCT parameters and coronary ischemia identified by QFR in patients with ICL. Additionally, we intended to determine the relationship between coronary stenosis derived from OCT and QFR.

## Materials and methods

### Study design and study population

This retrospective observational study consecutively identified patients with ICL undergoing OCT, who were admitted to the First Affiliated Hospital of Nanjing Medical

University from 1st June 2016–1st April 2020. The date of access to data for study purposes was from August 26th 2020 to September 1st 2021. The patients were included in this study if they met the diagnosis criteria of ICL in one or more target vessels, which were defined by visual estimation with diameter stenosis ranging from 50% to 90% and simultaneously assessed by OCT during coronary angiography. Subsequently, the exclusion criteria ruled out those with insufficient angiograms for calculating QFR. A total of 113 patients with 115 target vessels were enrolled. If the patient suffered from myocardial infarction, the assessment was only performed for the non-culprit lesions. Demographical, clinical, angiographic, and procedural data from the medical records of patients were obtained in this study. This study was approved by the ethics committee of the First Affiliated Hospital of Nanjing Medical University (2017-SR-328/2020-SR-359) and had been registered on Clinicaltrials.gov (NCT03229993). Considering the retrospective study design, the ethics committee approved the exemption of informed consent of each patient.

## QCA assessment

Quantitative angiographic analysis (QCA) regarding ICL was conducted by an accredited analyst blinded to the results of OCT and QFR through an automated software tool. The qualification was authorized by the National Coronary Intervention Training Center in Jiangsu Province Hospital (2020-15-03-0780). Analysis of QCA data including Minimal lumen area (MLA), Reference lumen area (RLA), and area stenosis (AS) were recorded.

## QFR assessment

A proprietary software package (Angio Plus, Pulse Medical Imaging Technology, Shanghai Co., Ltd., Shanghai, China) was used to calculate the QFR value. In total, two angiographic images (≥ 25° apart) were transferred to a computer which contained a QFR software, and conducted 3D reconstruction specific to the target vessel based on the previous description [8–10]. QFR was analyzed by an accredited technician who was blinded to the OCT data. The investigators were unblinded to the position of the lesions to ensure the comparison of QFR and OCT were at the same site. Within corresponding 2D acquisitions, the QFR software assisted in generating a 3D reconstruction for a single coronary vessel. The cut-off QFR value of no more than 0.80 indicated a significant ICL-associated ischemic burden. The QFR analyst was trained and certificated by Pulse Medical Imaging Technology (Shanghai) CO., LTD. (QFR-02180119).

## OCT image acquisition

OCT imaging was performed at target lesions with an FD-OCT system (Dragonfly™ OCT Intravascular Imaging, St. Jude Medical, Abbott Laboratory, United States). After positioning the Dragonfly™ OCT catheter to the distal end of the target lesion-automatic pull-back was performed to scan the entire target region. During the process of image acquisition, iso-osmolar iodixanol (Visipaque™ by GE Healthcare) was injected and filled the coronary artery, creating a blood-free condition. Lightlab imaging software was applied to do OCT image analysis. The OCT analyst had completed and obtained the certificate of CRT (Cardiovascular Research Technologies)-CSC (Chinese Society of Cardiology) Partnership International OCT Training Course.

## OCT assessment

Quantitative morphological assessments were conducted as follows [2]:

1. MLA was calculated as the cross-section for the smallest intraluminal area.

2. RLA was calculated as the reference cross-section with the largest lumen area within 10 mm proximal or distal to the MLA before any major side branch.

3. AS=(RLA-MLA)/RLA×100.

## Statistical analysis

SPSS version 23.0 (IBM Corp., Armonk, NY) was used to conduct statistical analysis. Categorical variables were presented with counts or percentages with the χ2 test or Fisher's exact test were used for comparisons. Continuous variables were presented with means ± standard deviations. The independent t-test was used to compare continuous variables between the two groups. A receiver-operating characteristic (ROC) curve analysis supported identifying the most suitable cut-off value to predict the QFR ≤ 0.8. The most suitable cut-off value was identified based on the highest sum of the sensitivity and specificity. A DeLong test compared the areas under the curve (AUC). Differences between area stenosis obtained with OCT and QFR were calculated and expressed in Bland-Altman plots. A two-sided p-value<0.05 was regarded as statistically significant.

## Results

### Baseline characteristics

Clinical registration data from 113 patients with 115 target vessels were collected, and a workflow diagram for this study was shown in Fig 1. Among the 113 patients, 32.7% were female, and the average age was (61.82 ± 10.26) years. The overall median QFR was 0.82 ± 0.10. Vessels with hemodynamic ischemia (n = 55) or without ischemia (n = 60) were based on the QFR cutoff value (QFR = 0.80). No complications were observed during the OCT procedure. The mean MLA and AS assessed by OCT were $3.11 \pm 1.54\,\mathrm{mm}^2$ and 68.4 ± 8.8%, respectively.

Table 1 showed demographic baseline and stenosis characteristics based on angiography, QCA, OCT, and QFR. Table 2 shows the characteristics of target vessels with ICL by QCA, OCT, and QFR according to their QFR values (≤0.80 or>0.80).

### Association between OCT and QFR measurements

OCT-measured MLA (r = 0.55, P < 0.01) and AS (r = 0.62 P < 0.01) were significant for predicting QFR values. The MLA-QFR and AS-QFR scatter plot graphs, as well as the regression analysis lines are shown in Fig 2.

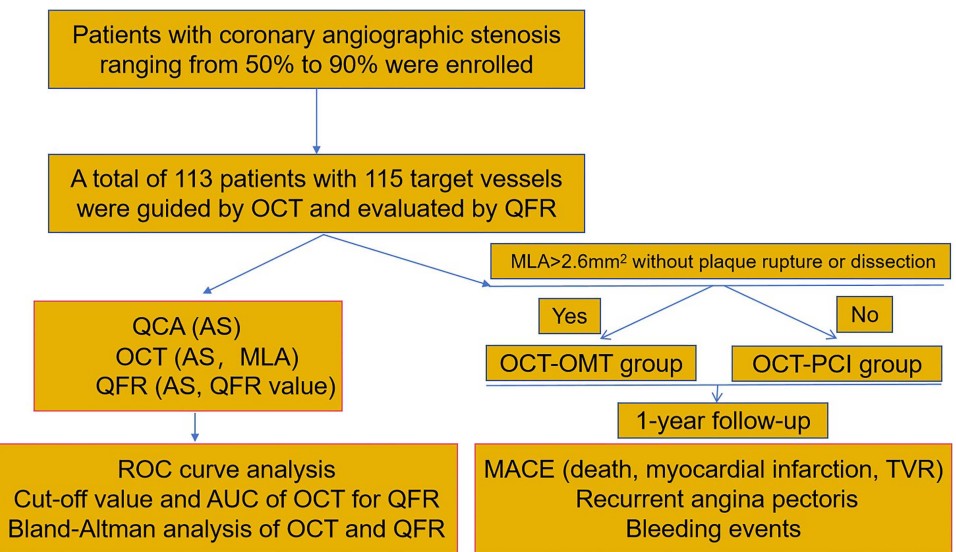

**Fig 1. The workflow diagram for the study.** AS, area stenosis; AUC, under the curve; MACE, major adverse cardiovascular events; MLA, minimal lumen area; OCT, optical coherence tomography; OMT, optimal medicine therapy; PCI, percutaneous coronary intervention; QFR, quantitative flow ratio; ROC, receiver-operating characteristic; TVR, target vessel revascularization.

**Table 1. Clinical and lesion characteristics by angiography, QCA, OCT and QFR.**

| Clinical characteristics (n = 113) | |
|---|---|
| Age (years) | 61.82 ± 10.26 |
| Male, n (%) | 76 (67.26%) |
| Hypertension, n (%) | 65 (57.52%) |
| Diabetes mellitus, n (%) | 26 (23.01%) |
| Clinical presentation | |
| Stable angina pectoris, n (%) | 13 (11.50%) |
| Unstable angina pectoris, n (%) | 91 (80.53%) |
| Acute myocardial infarction, n (%) | 9 (7.96%) |
| Angiographic characteristics (n = 115) | |
| Vessel investigated | |
| Left anterior descending artery, n (%) | 77 (66.96%) |
| Left circumflex artery, n (%) | 13 (11.30%) |
| Right coronary artery, n (%) | 25 (21.74%) |
| QCA characteristics (n = 115) | |
| Reference lumen area (mm$^2$) | 8.07 ± 3.61 |
| Minimal lumen area (mm$^2$) | 2.31 ± 1.22 |
| Area stenosis (%) | 71.6 ± 7.2 |
| OCT characteristics (n = 115) | |
| Reference lumen area (mm$^2$) | 9.84 ± 3.83 |
| Minimal lumen area (mm$^2$) | 3.11 ± 1.54 |
| Area stenosis (%) | 68.4 ± 8.8 |
| QFR characteristics (n = 115) | |
| QFR value | 0.82 ± 0.10 |
| Area stenosis (%) | 67.0 ± 7.2 |

Values are presented as n, n (%), or mean ± SD. OCT, optical coherence tomography; QCA, quantitative coronary angiography; QFR, quantitative flow ratio.

**Table 2. Differences in QCA and OCT characteristics based on QFR value.**

| | QFR ≤ 0.8 | QFR > 0.8 | *P* value |
|---|---|---|---|
| QCA characteristics (*n* = 115) | *n* = 55 | *n* = 60 | |
| Reference lumen area (mm$^2$) | 7.34 ± 3.16 | 8.73 ± 3.88 | 0.039 |
| Minimal lumen area (mm$^2$) | 1.79 ± 0.87 | 2.78 ± 1.31 | <0.001 |
| Area stenosis (%) | 75.6 ± 6.42 | 68.0 ± 5.96 | <0.001 |
| OCT characteristics (*n* = 115) | | | |
| Reference lumen area (mm$^2$) | 9.04 ± 3.26 | 10.57 ± 4.19 | 0.033 |
| Minimal lumen area (mm$^2$) | 2.31 ± 0.87 | 3.85 ± 1.66 | <0.001 |
| Area stenosis (%) | 74.0 ± 6.12 | 63.3 ± 7.73 | <0.001 |

Values are presented as mean ± SD. OCT, optical coherence tomography; QCA, quantitative coronary angiography; QFR, quantitative flow ratio.

Specific ROC curves for MLA and AS were calculated. ROC analysis revealed that OCT parameters, including MLA (AUC 0.83, 95% CI 0.75 to 0.90) and AS (AUC 0.87, 95% CI 0.81 to 0.94), had moderate diagnostic value for predicting QFR ≤ 0.8. The most appropriate cut-off values were 2.52 mm$^2$ for MLA (sensitivity 71%, specificity 85%) and 69% for AS

(sensitivity 84%, specificity 78%) (Fig 3). AS showed a better diagnostic value than MLA, while no statistical differences between the curves were shown after using the DeLong test.

The diagnostic value of MLA was high for identifying significant stenosis in right coronary artery. For the lesions located in the right coronary artery, the best cut-off values were 67% for AS (AUC 0.93, 95% CI 0.75–0.99, specificity 90%, sensitivity 86%) and 2.46 mm² for MLA (AUC 0.87, 95% CI 0.67–0.97, specificity 80%, sensitivity 93%). However, compared to the right coronary artery, the left coronary artery was associated with a relatively lower diagnostic performance in predicting the positive QFR value.

## Comparison of AS measured by OCT and QFR

Bland-Altman analysis for AS measured by OCT and QFR was shown in Fig 4a. The mean absolute differences and limits of agreement in AS measured with OCT and QFR were 1.4% and −12% to 9.2%, respectively. 7.8% (9/115) were out of the 95% limits of agreement; Fig 4b showed scatter plot graph and regression analysis line for AS measured with OCT and QFR. AS measured with QFR was found to significantly predict AS measured with OCT (r=0.79, p<0.001).

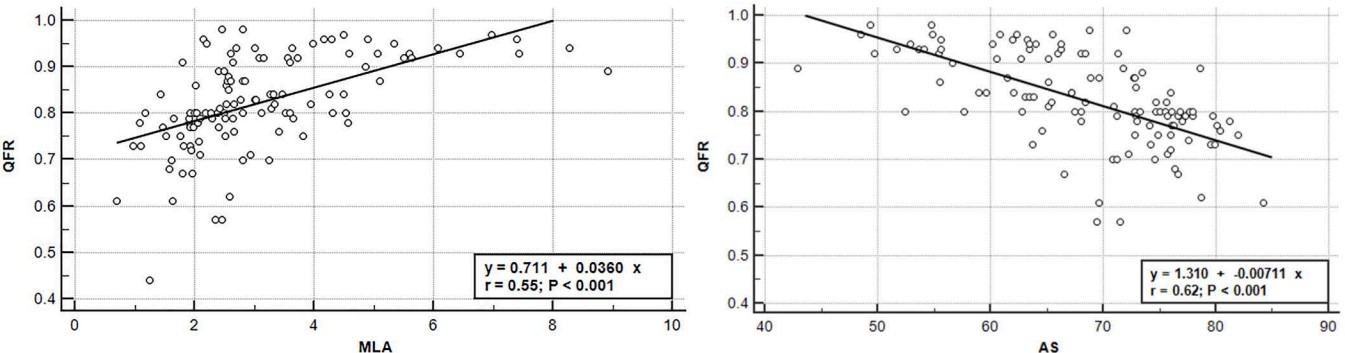

**Fig 2. Linear regression depicting the relationship between QFR values and OCT parameters (MLA, AS).** AS, area stenosis; MLA, minimal lumen area; OCT, optical coherence tomography; QFR, quantitative flow ratio.

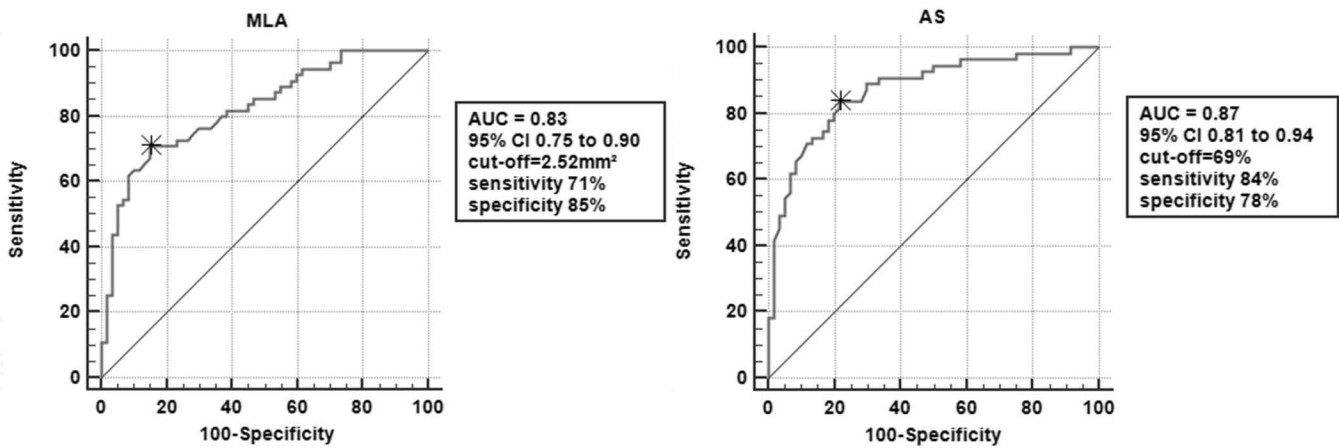

**Fig 3. Receiver-operating characteristic (ROC) curves for diagnostic accuracy of OCT-derived MLA and AS in predicting QFR≤0.8.** AS, area stenosis; MLA, minimal lumen area; OCT, optical coherence tomography; QFR, quantitative flow ratio.

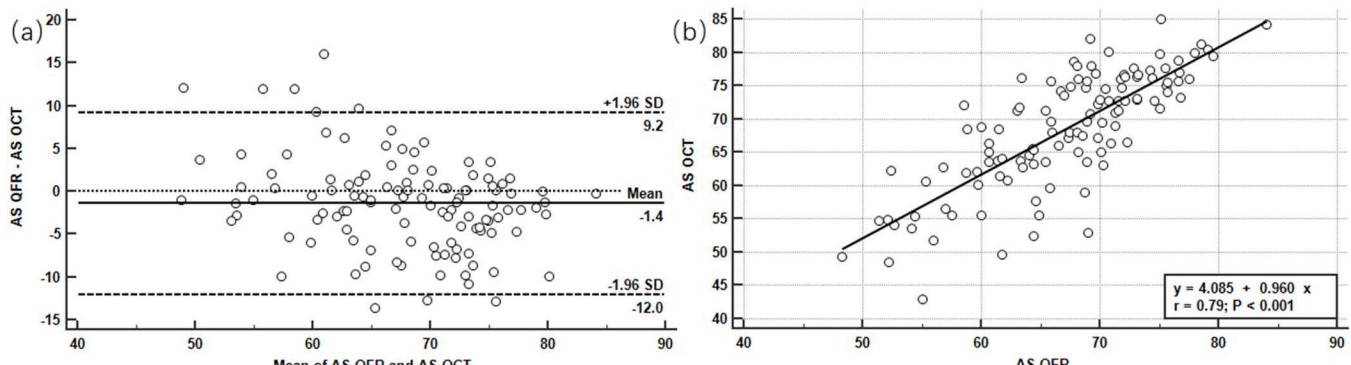

**Fig 4. (a) Bland-Altman plots showing the differences in AS between OCT and QFR; (b) Linear regression depicting the relationship between OCT-derived and QFR-derived AS.** AS, area stenosis; OCT, optical coherence tomography; QFR, quantitative flow ratio.

## Comparison of clinical characteristics and 1-year follow-up outcomes between OCT guided PCI and optimal medicine therapy (OMT)

In this study, OCT was performed on all patients. Patients with MLA > 2.6 mm$^2$, without plaque rupture or dissection [11], received optimal medical therapy (OMT) only. While OCT guided PCI was performed in the remaining patients. Overall, 46 patients (40.7%) were enrolled in the OCT guided OMT (OCT-OMT) group and 67 patients (59.3%) in the OCT guided PCI (OCT-PCI) group. There was no significant difference in age, gender, complications, medications or diagnosis between the two groups, while the persistent use ratio of dual antiplatelet drugs and Gensini score in the OCT-OMT group were significantly lower than those in the OCT-PCI group. At 1-year of follow-up, the incidence of major adverse cardiovascular events (MACE), including death, myocardial infarction and target vessel revascularization was 2.7% (3/113). There was no significant difference in the incidence of MACE, rehospitalization due to recurrent angina pectoris or major bleeding between the two groups. Whereas the OCT-PCI group was associated with a higher incidence of minor bleeding than the OCT-OMT group (Table 3).

## Discussion

The optimal management of patients presenting with ICL remains an unaddressed clinical need. Various adjunctive techniques have been adopted during the coronary angiography and visual estimation remains the primary way to assess ICL. Given the intrinsic limitations of visual estimation in guiding coronary revascularization, the application of FFR for functional evaluation has been considered an invaluable procedure to improve clinical outcomes [12,13]. Recently, several studies have confirmed that QFR with proven efficacy and reproducibility shows good diagnostic concordance with FFR [8–10]. This procedure does not require induction of hyperaemia or use of a pressure wire. This simple and non-invasive method holds promise for future clinical applications. OCT provides intravascular anatomical data and helps identify high risk ICL-related lesions. This study, for the first time, used QFR as a standard for demonstrating functionally severe stenosis and assessed the diagnostic value demonstrated by OCT for the identification of positive QFR. In addition, we aim to determine the association between AS obtained from QFR and AS obtained from OCT.

The main findings of this study are as follows: The established definition for intermediate coronary lesions is 50%−90% based on the latest European Society of Cardiology (ESC) Guidelines on Myocardial Revascularization. It is recommended that a comprehensive assessment of myocardial ischemia be performed unless there is > 90% diameter stenosis [14]. Several studies assessing the diagnostic value demonstrated by OCT for the identification of stenosis severity assessed by FFR described various optimal cut-off values with MLA (range, 1.6–2.5 mm$^2$) [3–7].

**Table 3. Comparison of clinical characteristics and 1-year follow-up outcomes.**

| | OCT guided OMT Group (N = 46) | OCT guided PCI Group (N = 67) | P value |
|---|---|---|---|
| **General information** | | | |
| Age, years | 60.9 ± 10.2 | 62.4 ± 10.3 | 0.44 |
| Male | 29 (63.0%) | 47 (70.1%) | 0.44 |
| Hypertension | 25 (59.5%) | 40 (58.9%) | 0.58 |
| Diabetes mellitus | 12 (21.6%) | 14 (17.9%) | 0.52 |
| **Medication** | | | |
| Aspirin* | 42 (89.2%) | 67 (100%) | 0.05 |
| P2Y12 receptor antagonist* | 24 (54.1%) | 67 (100%) | <0.0001 |
| Statins | 45 (97.3%) | 67 (100%) | 0.38 |
| Ezetimibe | 8 (13.5%) | 18 (23.2%) | 0.25 |
| β-blocker | 17 (37.8%) | 37 (53.6%) | 0.07 |
| ACEI or ARB/ANRI | 17 (40.5%) | 38 (60.7%) | 0.05 |
| calcium antagonist | 17 (37.8%) | 23 (37.5%) | 0.77 |
| Nitrates | 26 (62.2%) | 37 (57.1%) | 0.89 |
| Trimetazidine | 13 (27.0%) | 26 (30.4%) | 0.26 |
| **Diagnosis** | | | 0.81 |
| Angina pectoris | 42 (91.3%) | 62 (92.5%) | |
| Myocardial infarction | 4 (8.7%, Erosion) | 5 (7.5%) | |
| **Gensini score** | 17.6 ± 15.1 | 29.9 ± 19.6 | 0.0002 |
| **Adverse cardiovascular events (1-year follow-up)** | | | |
| Death | 0 | 0 | – |
| Myocardial infarction | 0 | 2 (3.0%) | 0.42 |
| Target vessel revascularization | 0 | 3 (4.5%) | 0.29 |
| Rehospitalization | 3 (6.5%) | 5 (7.5%) | 0.85 |
| **Bleeding events (1-year follow-up)** | | | |
| Major bleeding | 0 | 1 (1.5%) | 0.65 |
| Minor bleeding | 1 (2.2%) | 13 (19.4%) | 0.03 |

Note:

*Angina patients with stable lesions detected by OCT in the OMT group were switched to mono-antiplatelet therapy after receiving dual antiplatelet therapy for one month. ACEI, angiotensin converting enzyme inhibitor; ARB, angiotensin receptor inhibitor; OMT, optimal medical therapy; PCI, percutaneous coronary intervention.

The optimal cut-off value for AS in most studies is approximately 70%. According to our findings, MLA and AS showed moderate diagnostic efficiency in predicting the severity of functional stenosis assessed by QFR. We identified MLA of 2.52 mm$^2$ and AS of 69% to identify significant coronary stenosis, using QFR ≤ 0.8 as a standard, values that approach studies for predicting FFR ≤ 0.8 with OCT. Our study showed that the optimal cutoff value reaches similar values compared with previous FFR studies [3–7].

For the OCT parameters evaluated, AS showed better but not significant diagnostic efficiency in predicting QFR ≤ 0.8. This finding is in accordance with several previous results using FFR as a standard. AS takes into account the area of the reference vessel and balances the influence of lesion location and vessel diameter as much as possible, it can provide a more comprehensive assessment of lumen dimensions than MLA. Additionally, this study found a high diagnostic efficiency (AUC = 0.93) for AS in identifying QFR ≤ 0.8 in lesions located in the right coronary artery (RCA). The RCA data presented in this study cover limited lesions, so further investigations is needed to verify these results.

 

Our study suggested that OCT can be considered as an important complement to standard physiology, especially in some situations where physiologic information cannot be obtained. In fact, a recent study showed that OCT-guided PCI may reduce the incidence associated with the composite endpoint of MACE or significant angina compared with FFR-guided PCI [15]. OCT, a high-resolution intracavitary imaging technique, can not only measure lumen data but also identify vulnerable plaques and plaque anatomy and guide clinical decision making. In addition, OCT can guide stent implantation, identify stent-related complications with its high repeatability, and improve clinical prognosis [16].

However, our study identified only a modest diagnostic efficiency with limited sensitivity and specificity for demonstrating functionally severe stenosis. This suggests that the functional severity of coronary stenosis depends not only on the dimension of the coronary vessel lumen but also on other factors, including morphological details, the location of the stenosis, and the perfusion area of the vessels where the target stenosis located.

A further significant finding was that AS obtained by the QFR systems showed good correlation with that assessed by OCT. It is known that visual estimation can produce variability according to different observers and always overestimate the severity of stenosis [17–19]. Also, our previous study showed that 2D-QCA overestimates AS compared with OCT [20]. In fact, some studies have shown that conventional 2D-QCA measurements may be inaccurate in providing coronary stenosis information on lesions, especially in side branches [21–23]. It is known that visual estimation and 2D-QCA analysis can be inaccurate and unstable in measuring luminal stenosis and overestimate the severity of coronary stenosis, thereby leading to overtreatment and inappropriate clinical decisions. In our study, QFR, as a computational angiography-based modality without pressure wire and induced hyperemia, may be useful to provide lumen anatomical information in addition to its diagnostic performance or functional stenosis.

Our study had several limitations. First, this study was limited to a single-center retrospective study with a small sample size as a consequence of funding limitations, and additional large-scale multicenter studies are needed to confirm our findings. Second, although patients were enrolled consecutively, the exclusion criteria excluded lesions with insufficient angiograms, leading to potential selection bias. Third, although QFR has high concordance with FFR, recent studies suggest that microvascular resistance may decrease the diagnostic accuracy demonstrated by specific QFR for estimating FFR [24]. Failure to consider these factors could result in an inaccurate assessment of functional coronary stenosis. In addition, the current data are considered only hypothetical, as QFR has not been used as a standard to indicate physiologic ischemia and its effect on prognosis has not been determined. Recently, a prospective randomized trial (FAVOR III) compared the QFR-guided PCI and angiography-guided PCI in terms of clinical outcome as well as cost-effectiveness in evaluating patients with coronary artery disease [25].

## Conclusions

Our study explored the association between OCT and QFR and compared the advantages of each method. This study identified a moderate diagnostic accuracy of OCT for predicting functionally severe stenosis assessed by QFR. OCT can play a complementary role in guiding decision making in ICL by offering physiological information in addition to its typical value for morphological and anatomical assessment. QFR may offer a rapid and simple method evaluating area stenosis that reflects lumen anatomical information in coronary intermediate lesions.

## Supporting information

**S1 File. Clinical characteristics and follow-up results; baseline data of OCT and QFR.**
(XLSX)

## Acknowledgments

We would like to express our gratitude to all the participants and authors of this study.

## Author contributions

**Data curation:** Yuchen Zhang, Yang Yang.

**Investigation:** Peng Wu.

**Methodology:** Peng Wu.

**Writing – original draft:** Yuan Xue.

**Writing – review & editing:** Xin Lu, Haoyu Meng.

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
