## [Decision Letter · Decision Letter 0]

31 Jan 2025

PONE-D-24-51620Association between optical coherence tomography and functionally severe stenosis assessed by quantitative flow ratio in coronary intermediate lesionsPLOS ONE

Dear Dr. Lu,

Thank you for submitting your manuscript to PLOS ONE. After careful consideration, we feel that it has merit but does not fully meet PLOS ONE’s publication criteria as it currently stands. Therefore, we invite you to submit a revised version of the manuscript that addresses the points raised during the review process.

We look forward to receiving your revised manuscript.

Kind regards,

Giuseppe Filiberto Serraino, M.D., Ph.D.

Academic Editor

PLOS ONE

Journal Requirements:

 “National Natural Science Foundation of China (82400457)”

“All the authors declared no conflict of interest.”

4. In the online submission form, you indicated that [Data are available from the corresponding author upon reasonable request.].

Reviewers' comments:

Reviewer's Responses to Questions

**Comments to the Author**

1. Is the manuscript technically sound, and do the data support the conclusions?

Reviewer #1: Yes

Reviewer #2: Yes

2. Has the statistical analysis been performed appropriately and rigorously? 

Reviewer #1: Yes

Reviewer #2: Yes

3. Have the authors made all data underlying the findings in their manuscript fully available?

Reviewer #1: No

Reviewer #2: Yes

4. Is the manuscript presented in an intelligible fashion and written in standard English?

Reviewer #1: No

Reviewer #2: Yes

5. Review Comments to the Author

Reviewer #1: Review Comments

Title: Association between optical coherence tomography and functionally severe stenosis assessed by quantitative flow ratio in coronary intermediate lesions.

Background:

Thank you for inviting me to review this important manuscript. The authors have done well in studying a very important subject. They sought to investigate the association between OCT parameters (minimal lumen area and area stenosis) and functionally significant coronary stenosis identified by QFR (QRF 0.80 or less) in patients with intermediate coronary lesions. They found a good correlation between OCT parameters and QRF-determined severe stenosis. I have outlined my comments below.

Comments:

Abstract, introduction

1. Abstract (page 2): The authors should consider adding some baseline characteristics of the participants to the results session.

2. Introduction (page 3): The authors state, “However, the relationship between OCT and coronary stenosis with functional significance under the assessment of QRF has yet to be investigated.” This statement needs a bit more clarity.

3. The authors state, “This study centered on assessing the diagnostic value exhibited by OCT parameters in identifying coronary stenosis with functional significance under the assessment of QRF. This is not clear, and the objective needs to be more specific.

Methods

4. Study design and study population: The authors state, “Date of access to the data for study purposes was 26th August 2020.” – Does this mean they used one day to extract all the data? I think the period for data capturing should be stated.

5. The authors need to be clear on how the patients were selected. Were there any inclusion and exclusion criteria?

6. “This study had been registered in the net of Clinicaltrials.gov” – This needs grammatical correction. They also state the requirement for informed consent was waived. It must be stated which body waived it.

7. There are some grammatical errors in the methodology that need to be corrected.

Results

1. The authors should consider including the participants' sex in the baseline characteristics. What percentage were females?

2. The sample size of 113 patients with 115 lesions is relatively small.

3. The references to the tables and figures did not appear well in the text. The figures should be referred to appropriately in the text, for example, fig (a) on page 9 first line should be fig (4a)

4. Figures and tables: the explanation of the abbreviations should be written alphabetically. In table 3, ‘Ezemab’ should be changed to ezetimibe.

5. The authors state on page 9, paragraph 2, “Upon stenosis being divided according to lesion location in order to assess the diagnostic performance at different vessels, the diagnostic value exhibited by MLA for the identification of significant stenosis for the right coronary artery was heightened.” Rephrasing this will bring some clarity.

6. On page 10, the authors state that OCT was performed for ALL the patients. Meanwhile, 37 patients (40%) were enrolled in the OCT-guided OMT group and 56 patients (60%) in the OCT-guided PCI (OCT-PCI) group. What happened to the rest of the 113 participants?

7. Figure 1 (the flow chart): The authors stated in the methods (page 4) that patients who did not have good angiograms for QRF analysis were excluded from the analysis. However, the figure does not indicate how many patients were excluded based on this reason.

Discussion

1. Some grammatical and typographical errors need to be corrected. Eg ‘ASacquired’ should be ‘AS-acquired’ (page 11, last sentence); OCT-guided PCI instead of OCT-guidance PCI (page 13); FFR-guided PCI instead of FFR-guidance PCI (page 13).

2. The authors need to expand the limitations session, addressing, for example, the small sample size, confounders, and generalizability.

Acknowledgement – Please change ‘my’ to ‘our’.

Reviewer #2: Comments:

Review PLOS ONE, Manuscript: 24 51620

Association between optical coherence tomography and functionally severe stenosis assessed by quantitative flow ratio in coronary intermediate lesions

Article Type: Research Article.

Objective: To investigate the association between optical coherence tomography (OCT) parameters and coronary functionally significant stenosis identified by quantitative flow ratio (QFR) in patients with intermediate coronary lesion (ICL).

General comments: An interesting paper.

Strengths:

A novel topic.

Weaknesses:

Some areas require much more detail.

TITLE:

Comment 1: I think rephrasing the Title along the lines of “Accuracy/relationship of optical coherence tomography to detect/quantify functionally severe stenosis assessed/evaluated by quantitative flow ratio in intermediate coronary lesions” would better reflect the research as I understand it.

I would try to describe this is terms of a PICOS statement

P = Patient, problem or population = patients with intermediate coronary lesion (ICL)

I = Intervention = optical coherence tomography (OCT)

C = Comparison, control or comparator = coronary artery stenosis evaluated by quantitative flow ratio (QFR)

O = Outcome(s) = Accuracy, sensitivity, specificity

S = Study design = a retrospective observational study of diagnostic test accuracy (DTA)

ABSTRACT:

Appropriate.

Keywords: None.

Abbreviations: None.

INTRODUCTION: Appropriate.

The rationale for the study, in the context of what is already known is provided.

The purpose of the study is stated.

MATERIALS and METHODS:

Ethics review. Was the need for informed consent waived?

A retrospective observational study.

A single institution study.

This study had been registered in the net of Clinicaltrials.gov

Comment 2: Did participants form a consecutive, random, or convenience series?

Comment 3: What was the Intended sample size and how it was determined?

Comment 4: Please state explicitly the inclusion and exclusion criteria.

University hospital setting.

Date of study given.

Comment 5: Please describe the study population in terms of background?

QCA assessment

Comment 6: Please provide more technical specifications of the QCA technique, units, cutoffs, etc., and details of the training, and expertise of the persons executing and reading this test.

QFR assessment

Comment 7: Please provide units, cutoffs, etc., and details of the training, and expertise of the persons executing and reading this test.

Comment 8: Please provide details of the training, and expertise of the persons executing and reading this test.

Statistical analyses: Appropriate.

Statistical methods and analytical approach and statistical software that was used for data analysis given and appropriate.

RESULTS:

Association between OCT measurements and QFR measurements: Appropriate

Comparison of AS measured by OCT and QFR: Appropriate

Comparison of clinical characteristics and 1-year follow-up outcomes between OCT guided PCI and optimal medicine therapy (OMT): Appropriate

DISCUSSION

Thorough review of the literature.

The study’s strengths are discussed

The study’s limitations are discussed

Conclusions: Appropriate

Research funded details provided.

Author contributions provided

There are no conflicts of interest

REFERENCES: Appropriate.

Tables: Appropriate.

Table Legends: Appropriate.

Figures: Appropriate.

Figure Legends: Appropriate.

6. PLOS authors have the option to publish the peer review history of their article (what does this mean? ). If published, this will include your full peer review and any attached files.

**Do you want your identity to be public for this peer review?** For information about this choice, including consent withdrawal, please see our Privacy Policy .

Reviewer #1: **Yes: ** Francis Agyekum

Reviewer #2: No

---

## [Author Response · Author response to Decision Letter 1]

4 Mar 2025

Dear Editor:

Thank you very much for your letter and reviewers' comments concerning our manuscript entitled “Association between optical coherence tomography and functionally severe stenosis assessed by quantitative flow ratio in coronary intermediate lesions” (ID: PONE-D-24-51620). Those comments are all valuable and helpful for revising and improving our paper, as well as the important guiding significance to our researches. We have studied the comments carefully and have made correction which we hope meet with approval. Revised portion are highlighted in red in the revised manuscript. The main revision in the paper and the responses to the reviewers' comments are as following:

Comments:

Abstract, introduction

1. Abstract (page 2): The authors should consider adding some baseline characteristics of the participants to the results session.

Response: Thank you for your good advice. We have incorporated a comprehensive overview of the baseline characteristics of the participants in the revised manuscript.

2. Introduction (page 3): The authors state, “However, the relationship between OCT and coronary stenosis with functional significance under the assessment of QRF has yet to be investigated.” This statement needs a bit more clarity.

Response: We appreciate it very much for this good suggestion. We have added this in the updated manuscript with red highlighting sentences.

3. The authors state, “This study centered on assessing the diagnostic value exhibited by OCT parameters in identifying coronary stenosis with functional significance under the assessment of QRF. This is not clear, and the objective needs to be more specific.

Response: Thank you for your carefully review of this paper, and we have done it according to your ideas.

Methods

4. Study design and study population: The authors state, “Date of access to the data for study purposes was 26th August 2020.” – Does this mean they used one day to extract all the data? I think the period for data capturing should be stated.

Response: Thank you for your comments. We are very sorry to have confused you. The initial access date for data utilization for research purposes was August 26, 2020. We have amended the pertinent statement in accordance with the requirements.

5. The authors need to be clear on how the patients were selected. Were there any inclusion and exclusion criteria?

Response: Thank you for pointing this out. We have supplemented the inclusion and exclusion criteria in the updated manuscript .

6. “This study had been registered in the net of Clinicaltrials.gov” – This needs grammatical correction. They also state the requirement for informed consent was waived. It must be stated which body waived it.

Response: Thank you very much for pointing this error out. We are sorry for this grammatical problem and have corrected it according to your suggestion.

7. There are some grammatical errors in the methodology that need to be corrected.

Response: Thank you for your carefully review of this paper. We are sorry for the grammar and vocabulary mistakes and have tried our best to correct those errors according to your suggestion. In addition, the manuscript has been revised by a professional English editor who is a native speaker.

Results

1. The authors should consider including the participants' sex in the baseline characteristics. What percentage were females?

Response: Thank you for pointing this out. Among the 113 patients, 32.7% were female.

2. The sample size of 113 patients with 115 lesions is relatively small.

Response: Thank you for raising the issue regarding the sample size of our study. We acknowledge that a cohort of 113 patients with 115 lesions may be considered limited; however, this number was carefully determined based on the available resources and feasibility within the context of our research. To ensure the reliability and validity of our findings, we utilized rigorous statistical methods. In future studies, we plan to explore expanding the sample size to further validate our results. Furthermore, this limitation is elaborated upon in the limitations section.

3. The references to the tables and figures did not appear well in the text. The figures should be referred to appropriately in the text, for example, fig (a) on page 9 first line should be fig (4a).

Response: Thank you for your carefully review of this paper. The requested modifications have been implemented as specified. All citations of tables and figures have been meticulously reviewed and updated to accurately reflect the corresponding elements presented.

4. Figures and tables: the explanation of the abbreviations should be written alphabetically. In table 3, ‘Ezemab’ should be changed to ezetimibe.

Response: Thank you for pointing this out. We have corrected this error in the updated manuscript with red highlighting sentences.

5. The authors state on page 9, paragraph 2, “Upon stenosis being divided according to lesion location in order to assess the diagnostic performance at different vessels, the diagnostic value exhibited by MLA for the identification of significant stenosis for the right coronary artery was heightened.” Rephrasing this will bring some clarity.

Response: Thank you for your valuable counsel. The sentence has been revised in accordance with your request. The updated phrasing now clearly indicates that the authors categorized stenosis based on lesion location and discovered that the diagnostic value of MLA for identifying significant stenosis in the right coronary artery was enhanced.

6. On page 10, the authors state that OCT was performed for ALL the patients. Meanwhile, 37 patients (40%) were enrolled in the OCT-guided OMT group and 56 patients (60%) in the OCT-guided PCI (OCT-PCI) group. What happened to the rest of the 113 participants?

Response: Thank you very much for pointing this error out. We are sorry for this problem and have corrected it in the updated manuscript with red highlighting sentences. Overall, 46 patients (40%) were enrolled in the OCT guided OMT (OCT-OMT) group and 67 patients (60%) in the OCT guided PCI (OCT-PCI) group.

7. Figure 1 (the flow chart): The authors stated in the methods (page 4) that patients who did not have good angiograms for QRF analysis were excluded from the analysis. However, the figure does not indicate how many patients were excluded based on this reason.

Response: We appreciate it very much for this good suggestion, and we have done it according to your ideas. The flow chart has been revised. The updated version now includes the number of patients excluded due to poor angiograms for QRF analysis, providing a clearer understanding of the patient selection process.

Discussion

1. Some grammatical and typographical errors need to be corrected. Eg ‘ASacquired’ should be ‘AS-acquired’ (page 11, last sentence); OCT-guided PCI instead of OCT-guidance PCI (page 13); FFR-guided PCI instead of FFR-guidance PCI (page 13).

Response: We have corrected the grammatical and typographical errors as your instructions.

1. The authors need to expand the limitations session, addressing, for example, the small sample size, confounders, and generalizability.

Response: We have revised the limitations section as requested, addressing the small sample size, potential confounders, and the generalizability of the findings. The revised content provides a more comprehensive assessment of the study's constraints.

Acknowledgement – Please change ‘my’ to ‘our’.

Response: Thank you for pointing this out. We have corrected it as requested.

Thank you again for your positive and constructive comments and suggestions on our manuscript.

Reviewer #2: Comments:

Comment 1: I think rephrasing the Title along the lines of “Accuracy/relationship of optical coherence tomography to detect/quantify functionally severe stenosis assessed/evaluated by quantitative flow ratio in intermediate coronary lesions” would better reflect the research as I understand it.

Response: We deeply appreciate your suggestion on refining the title. After discussing among the team, we've decided to keep the original title as it aligns well with our research focus. Thank you for your understanding.

Comment 2: Did participants form a consecutive, random, or convenience series?

Response: Thank you for your carefully review of this paper. This retrospective observational study consecutively identified patients with ICL undergoing OCT, who were admitted to the First Affiliated Hospital of Nanjing Medical University from 1st June 2016 to 1st April 2020. We have added this in the updated Materials and Methods section.

Comment 3: What was the Intended sample size and how it was determined?

Response: In response to your inquiry regarding the intended sample size, we wish to clarify that, due to the retrospective nature of our study, we have endeavored to collect as many eligible cases as possible. The final sample size, which we believe is adequate for our analysis, consists of a total of 113 cases meeting the study criteria. Looking ahead, we acknowledge the importance of increasing our sample size to further strengthen our findings. Consequently, in the next phase of our research, we plan to expand the sample size to enhance the robustness and generalizability of our results.

Comment 4: Please state explicitly the inclusion and exclusion criteria.

Response: Thank you for pointing this out. We have supplemented the inclusion and exclusion criteria in the updated manuscript.

Comment 5: Please describe the study population in terms of background? QCA assessment

Response: Thank you for pointing this out. We have now included the requested information in the updated manuscript.

Comment 6: Please provide more technical specifications of the QCA technique, units, cutoffs, etc., and details of the training, and expertise of the persons executing and reading this test. QFR assessment

Response: We appreciate your valuable feedback. In response, we have revised the manuscript to incorporate detailed technical specifications of the QCA technique, including units of measurement, cut-off values, and critical technical parameters. Furthermore, we have elaborated on the qualifications and expertise of the personnel involved in conducting and interpreting the QFR assessments, emphasizing their extensive training and experience in this specialized field.

Comment 7: Please provide units, cutoffs, etc., and details of the training, and expertise of the persons executing and reading this test.

Response: Thank you for your good suggestion. The manuscript now incorporates specific units of measurement and cut-off values utilized in the test. With respect to the personnel responsible for executing and interpreting the test, they have undergone rigorous training in the relevant field and possess extensive expertise. Detailed information regarding their training and qualifications has been included in the Materials and Methods section. We trust that these revisions adequately address your concerns.

Comment 8: Please provide details of the training, and expertise of the persons executing and reading this test.

Response: Thank you for your comment. We have updated the manuscript to provide comprehensive details on the training and expertise of the persons executing and reading the test. They have undergone extensive specialized training, hold relevant professional certifications, and possess significant years of experience in the field. This ensures the accuracy and reliability of our test results. We hope this information meets your requirements.

We appreciate your elaborate efforts in reviewing. Thank you very much!

We tried our best to improve the manuscript and made some changes in the manuscript. These changes will not influence the content and framework of the paper. And here we did not list the changes but highlighted in red in the revised manuscript. We appreciate for Editors/Reviewers' warm work earnestly, and hope that the revision will meet with approval.

The authors have declared that no competing interests exist.

Once again, thank you very much for your comments and suggestions.

Sincerely yours,

Xin Lu and Haoyu Meng

Feb 23, 2025

Funding: Project supported by the National Natural Science Foundation of China (82400457), the Natural Science Foundation of Jiangsu Province (BK20210966) and the Clinical project of Jiangsu Provincial People's Hospital (303103513BA20). The funders had no role in study design, data collection and analysis, decision to publish, or preparation of the manuscript.

---

## [Decision Letter · Decision Letter 1]

2 May 2025

Association between optical coherence tomography and functionally severe stenosis assessed by quantitative flow ratio in coronary intermediate lesions

PONE-D-24-51620R1

Dear Dr. Lu,

We’re pleased to inform you that your manuscript has been judged scientifically suitable for publication and will be formally accepted for publication once it meets all outstanding technical requirements.

Kind regards,

Giuseppe Filiberto Serraino, M.D., Ph.D.

Academic Editor

PLOS ONE

Additional Editor Comments (optional):

Reviewers' comments:

Reviewer's Responses to Questions

**Comments to the Author**

1. If the authors have adequately addressed your comments raised in a previous round of review and you feel that this manuscript is now acceptable for publication, you may indicate that here to bypass the “Comments to the Author” section, enter your conflict of interest statement in the “Confidential to Editor” section, and submit your "Accept" recommendation.

Reviewer #3: All comments have been addressed

2. Is the manuscript technically sound, and do the data support the conclusions?

Reviewer #3: Yes

3. Has the statistical analysis been performed appropriately and rigorously? 

Reviewer #3: Yes

4. Have the authors made all data underlying the findings in their manuscript fully available?

Reviewer #3: Yes

5. Is the manuscript presented in an intelligible fashion and written in standard English?

Reviewer #3: Yes

6. Review Comments to the Author

Reviewer #3: Dear Authors,

I have reviewed the revised version of your manuscript. I would like to commend you for your diligent efforts in addressing the previous comments. With the revisions made, the manuscript has become significantly more valuable and informative. Congratulations on your thorough work.

Best regards

7. PLOS authors have the option to publish the peer review history of their article (what does this mean? ). If published, this will include your full peer review and any attached files.

**Do you want your identity to be public for this peer review?** For information about this choice, including consent withdrawal, please see our Privacy Policy .

Reviewer #3: **Yes: ** FERIT BOYUK, MD

---

## [Editor Report · Acceptance letter]

PONE-D-24-51620R1

PLOS ONE

Dear Dr. Lu,

I'm pleased to inform you that your manuscript has been deemed suitable for publication in PLOS ONE. Congratulations! Your manuscript is now being handed over to our production team.

Kind regards,

on behalf of

Professor Giuseppe Filiberto Serraino

Academic Editor

PLOS ONE